# A comprehensive, mechanistically detailed, and executable model of the cell division cycle in *Saccharomyces cerevisiae*

Ulrike Münzner[1,2], Edda Klipp [1] & Marcus Krantz[1]

Understanding how cellular functions emerge from the underlying molecular mechanisms is a key challenge in biology. This will require computational models, whose predictive power is expected to increase with coverage and precision of formulation. Genome-scale models revolutionised the metabolic field and made the first whole-cell model possible. However, the lack of genome-scale models of signalling networks blocks the development of eukaryotic whole-cell models. Here, we present a comprehensive mechanistic model of the molecular network that controls the cell division cycle in *Saccharomyces cerevisiae*. We use rxncon, the reaction-contingency language, to neutralise the scalability issues preventing formulation, visualisation and simulation of signalling networks at the genome-scale. We use parameter-free modelling to validate the network and to predict genotype-to-phenotype relationships down to residue resolution. This mechanistic genome-scale model offers a new perspective on eukaryotic cell cycle control, and opens up for similar models—and eventually whole-cell models—of human cells.

[1] Humboldt-Universität zu Berlin, Institute of Biology, Theoretical Biophysics, Berlin 10099, Germany. [2] Bioinformatics Center, Institute for Chemical Research, Kyoto University, Kyoto 611-0011, Japan. Correspondence and requests for materials should be addressed to M.K. (email: marcus.krantz@rxncon.org)

Computational models provide powerful tools to study biological systems[1]. In particular, mechanistic models that explain cellular functions and phenotypes from molecular events are powerful tools to assemble knowledge into understanding. These models combine three functions: as integrated and internally consistent knowledge bases, as scaffolds for integration, analysis and interpretation of data, and as executable models. Their value arguably increases the more mechanistically detailed and comprehensive they are, culminating in the genome-scale mechanistic models of metabolism and the whole-cell model of *Mycoplasma genitalium*[2–4]. These models can be used to explain and predict perturbation responses and genotype-to-phenotype relationships, and whole-cell models have the potential to revolutionise biology, biotechnology and biomedicine. However, to realise this potential, we must be able to build mechanistic genome-scale models of all cellular processes.

This has proven especially challenging for the cellular networks that process information (reviewed in refs. [5,6]). These networks encode information primarily through reversible state changes in their components, such as bonds or covalent modifications. Typically, these components interact with multiple partners and may be modified at multiple residues, and most of these bonds and/or modifications are not mutually exclusive. Consequently, there is a one-to-many relationship between empirical observables (elemental states; Fig. 1a), such as a specific bond or the modification status at a specific residue, and the possible configurations of the components (microstates). This leads to problems in most classical modelling formalism, where the resolution difference leads either to a loss of mechanistic detail (e.g. component level modelling) or to the combinatorial complexity (microstate modelling). The solution is to use a formalism with adaptive resolution, such as rule-based modelling languages (RBMLs)[7,8], the Entity Relationship diagrams of the Systems Biology Graphical Notation (SBGN-ER)[9] or rxncon, the reaction-contingency language[10] (reviewed in refs. [11,12]). However, the potential of these formalisms has yet to be realised in a comprehensive mechanistic model of a eukaryotic signalling system.

The eukaryotic cell division cycle (CDC) may be the most interesting—both medically and philosophically—of these systems, as it constitutes the very core of life as we know it. The CDC is best understood in baker's yeast, *Saccharomyces cerevisiae*: the rise and fall in activity of a single cyclin-dependent kinase (CDK) suffices to drive the replication of DNA, nuclear division (including duplication and separation of the spindle pole body (SPB, the yeast centrosome)) and cell morphology and division[13,14]. The molecular basis of cell-cycle control has been studied since the 1970-ies with experimental[15] as well as computational[16–19] methods. However, even the largest executable models are far from genome-scale[19], and the exquisitely detailed map created in the process description diagram language (SBGN-PD[9]) cannot be executed[20]. To realise the potential of a mechanistic genome-scale model, we need to combine the features of all three efforts: the mechanistic precision in the molecular biology, the scope of the comprehensive maps and the executability of the mathematical models.

Here, we present a mechanistic, executable and genome-scale model based on the network that controls and executes the cell division cycle in baker's yeast, *S. cerevisiae*. We chose to build this model using rxncon (Fig. 1), as it (similarly to RBMLs) has the adaptive resolution required to reconcile the necessary scalability and precision, and as a rxncon network (in contrast to a rule-based model (RBM)) can be compiled into and simulated as a parameter-free bipartite Boolean model (bBM)[21]. This scalable simulation method enables qualitative simulation of the cell-cycle network, which is too large and/or has too many unknown parameter values (or truth tables) to be simulated with classical methods. We use bBM simulation to validate the wild-type model, prompting us to introduce a number of gap-filling changes to create a functional cell-cycle model. Interestingly, we find that the control network consists of three distinct regulatory modules that control and communicate via three independent replication cycles: DNA replication, SPB duplication and nuclear division and cell division, and that only a hybrid model including both parts can mechanistically explain cell-cycle control. Finally, we benchmark the model on a set of 85 mutants, capturing 62/85 phenotypes. Taken together, we show that it is possible to build, visualise and simulate mechanistic models of signal transduction systems at the genome-scale, and that system level function can be predicted from the level of molecular mechanisms without parametrisation or model training.

## Results

**A comprehensive, mechanistic and executable knowledge base.** We compiled literature knowledge into a mechanistically detailed model of cell division cycle control in baker's yeast (Fig. 2a, Supplementary Data 1, Supplementary Data 2, Supplementary Figure 1). We derived the knowledge from in-depth literature curation[22–55] (see Supplementary Table 1 for complete list of model references) and formalised it in a rxncon model without any fitting or model optimisation. The model includes regulated expression and degradation of proteins, assembly and regulation of the cyclin-dependent protein kinases Cdc28 (Cdk1) and Pho85 (Cdk5), and the regulation of DNA replication, SPB duplication and nuclear division, and cell polarity and morphogenesis. Each of these processes is defined—as far as empirical knowledge allows—down to the role of specific modifications and bonds at particular residues and domains. This molecular reaction network (MRN) accounts for 357 unique components. This number includes 229 proteins, and the genes and mRNAs of the 44 proteins for which we consider regulated expression and/or degradation. The model also includes 7 multimeric protein complexes (e.g. APC/C), 3 chromosomal features (e.g. the origins of replication), 29 kinase and phosphatase activities as target specific enzymes that remain to be mapped on one or more gene products (e.g. Sfi1PPT as an unidentified phosphatase (PPT) of Sfi1), and one small molecule: phosphatidylinositol (PI). These components take part in 790 elemental reactions that produce and consume 1238 elemental states, and that are regulated by 598 contingencies—several of which correspond to combinations of elemental states.

In addition to the MRN, we encode a coarse-grained model (CGM) of DNA replication, SPB duplication and the morphological cell cycle in 12 macroscopic reactions and 12 macroscopic states (Fig. 2b, c). The macroscopic reactions respond to a series of inputs that constitutes outputs (observables) in the MRN. Conversely, the CGM outputs act as inputs to the MRN. The entire regulatory logic is encoded in a single hybrid rxncon model, encompassing 802 reactions and 972 lines of contingencies. The contingencies connect the MRN and CGM to each other. The contingencies also link to the five external inputs we consider to perturb the system allowing for further analysis: nutrients, pheromone, hydroxyurea (HU), latrunculin A (LatA) and nocodazole (NOC). The biology and implementation is described in detail in the extensive Supplementary Methods, where the network is divided into thirty smaller modules that are described and visualised individually (Supplementary Figures 2–31; Supplementary Methods). The model accounts for all components to which we could assign a mechanistic function in the control of cell-cycle division and, hence, it constitutes a first draft of a genome-scale mechanistic model (GSM) of eukaryotic cell division.

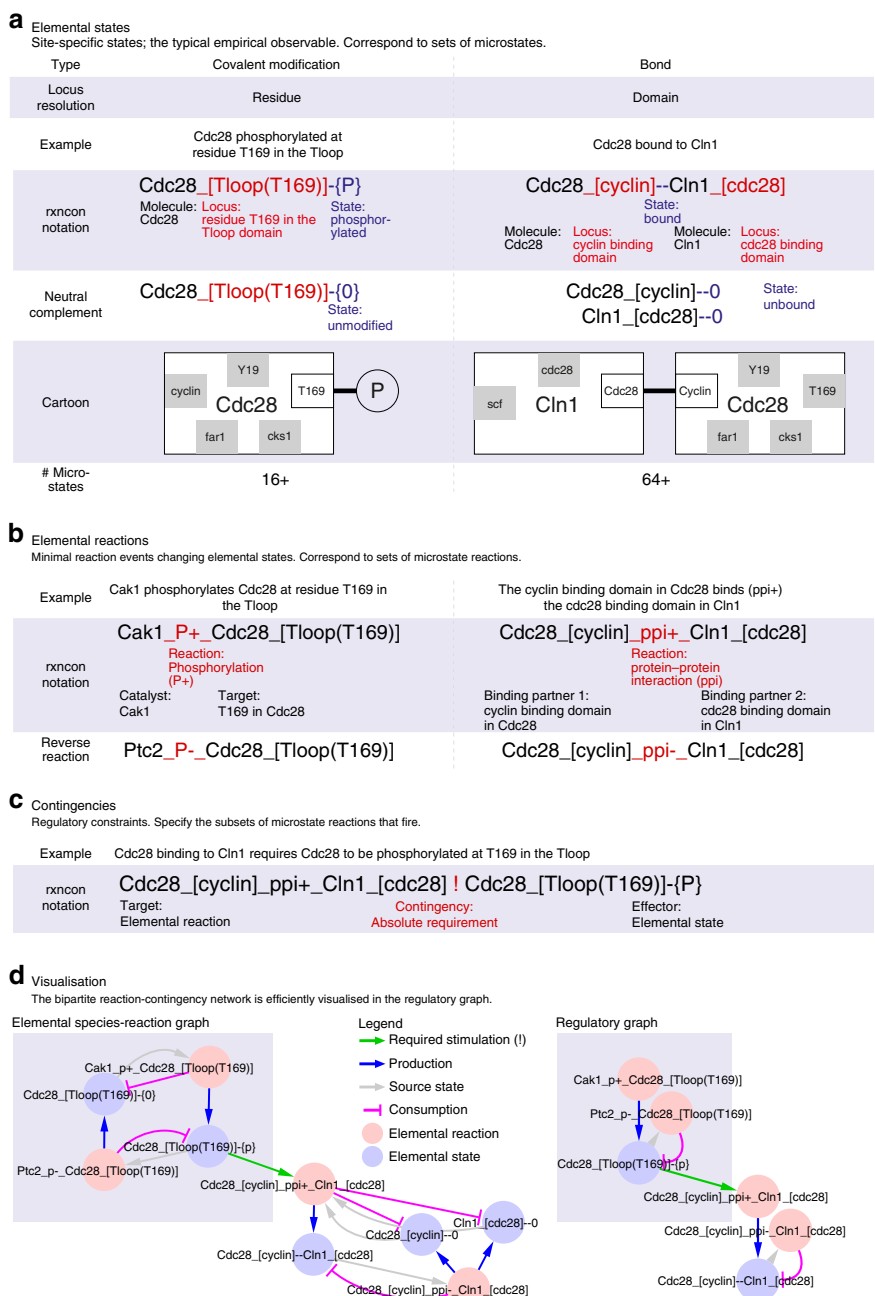

**Fig. 1** The reaction-contingency (rxncon) language employs three essential components: elemental states, elemental reactions and contingencies. The regulatory graph efficiently represents the regulatory structure of the network. **a** Elemental states are empirical observables: signalling molecules encode information through site-specific state changes, i.e. covalent modification of specific residues or bonds between specific domains. All elemental states at a single locus are mutually exclusive with each other and their neutral complements (unmodified and unbound, respectively). However, each elemental state only defines the state at a single locus (two for bonds; cartoon: white boxes), leading to a one-to-many relationship between the (empirically measured) elemental states and the (inferred) microstates. **b** Elemental reactions are indivisible reaction events defined in terms of elemental states. They are only defined in terms of the elemental states that are produced, consumed, synthesised or degraded, similar to the reaction centre of a rule in RBMs, and hence have a one-to-many relationship to microstate reactions. **c** The contingencies define constraints on an elemental reaction in terms of elemental states (or inputs) that do not change through the reaction. Hence, the contingencies reduce the number of valid microstate reactions, and correspond to the reaction contexts of RBMs. Boolean contingencies, employing AND, OR and NOT gates, can specify arbitrarily detailed constraints down to microstates as necessary. Due to this adaptive resolution, rxncon models can faithfully mirror the empirical knowledge. **d** The regulatory graph (right) provides a compact representation of the regulatory structure of the rxncon network. It is a simplified version of the elemental species-reaction graph (left), leaving out the neutral states to reduce graph complexity, to remove non-informative cycles, and to improve readability. Both graphs are bipartite, with two types of nodes and two types of edges: reaction-to-state (reaction) edges show the effect of each reaction on its source and product states, and state-to-reaction (contingency and source state) edges the impact of states on reactions. The bipartite nature of the graph highlights the requirement for both reactions and contingencies for information transfer through the network, as both types of edges are necessary to traverse the graph

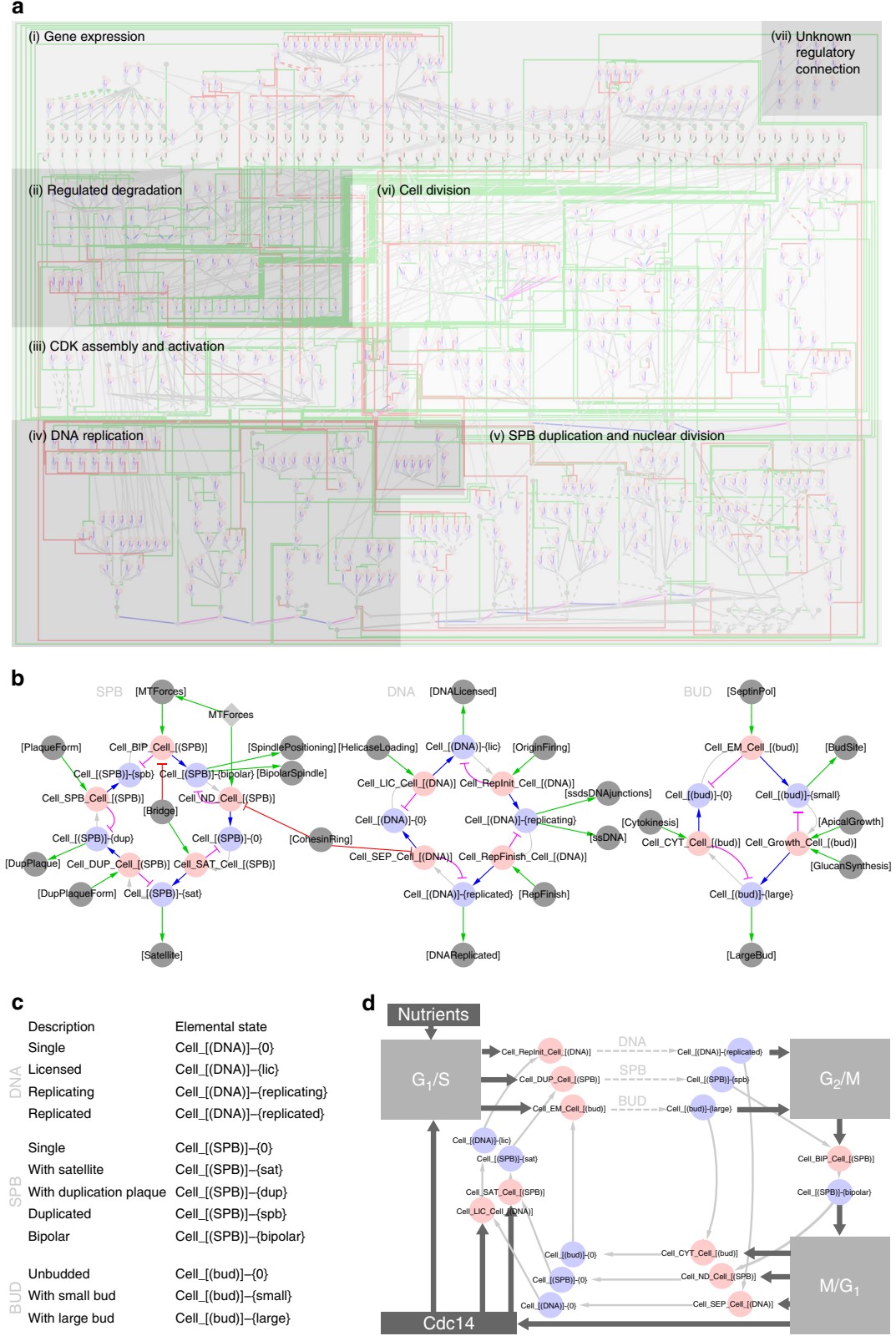

**Building a mechanistic knowledge base at the genome scale.** The GSM primarily constitutes a biological knowledge, which can be compiled into a mathematical model. In an iterative workflow[56], we extracted and evaluated empirical information, and formalised it as elemental reactions (e.g. Cdc28 and Cln1 bind via their cyclin and cdc28 binding domains, respectively; Fig. 1b) and contingencies (e.g. Cdc28 binds to Cln1 only if Cdc28 is phosphorylated at T169; Fig. 1c). This type of mechanistic model puts high demands on data coverage and quality. We identified knowledge gaps, which forced specific gap-filling assumptions to

**Fig. 2** The molecular architecture of the CDC control network. **a** Bird's eye view of the complete network (see Supplementary Figure 1 for a readable poster-sized version). The mechanistic reaction network (MRN) consists of seven parts: (i) Gene expression, (ii) regulated degradation, (iii) CDK assembly and activation, (iv) DNA replication, (v) SPB duplication and nuclear division, (vi) cell division and (vii) a set of reactions involving cell-cycle components but without (known) regulatory connections to the main network. The outputs of the MRN in modules (iv)–(vi) determine the progression of the coarse-grained model (CGM). **b** The CGM is encoded as twelve reactions and twelve states, divided into three distinct cycles of sequential transitions: (left) SPB duplication and nuclear division, (middle) DNA replication and (right) Cell division. The MRN and CGM parts communicate via an input/output layer (grey nodes). **c** Verbal equivalents of the rxncon states used to encode the DNA, SPB and BUD cycles in **b** and **d**. **d** Schematic representation of the network architecture. Three independent replication cycles are regulated by three disjunct regulatory modules gating the $G_1/S$, $G_2/M$ and $M/G_1$ transitions. After cytokinesis, the activity of Cdc14 (and APC/C) is necessary to reset the state of the cell-cycle network and allow DNA licensing and SPB bridge formation. At $G_1/S$, Cln1/2 and Clb5/6 activity are necessary to initiate and drive DNA replication, SPB duplication and Bud formation, dependent on sufficient nutrients. At $G_2/M$, DNA replication must be finished and Bud size and/or morphology sufficient to allow Clb1/2 expression and activity, which drives the cell into mitosis. At $M/G_1$, The spindle must be under tension and aligned, requiring the proper state in each of the three global cycles. The combined activity of Cdc14 and APC/C resets the control network, triggering cytokinesis and resetting the three replication cycles

create a fully connected model (Supplementary Data 1). In addition, we made the general assumptions that all reversible covalent modifications and all synthesised components are turned over. This lead us to introduce a set of 28 undefined phosphatase activities to compensate for the fact that these reaction types are understudied[10]. These assumptions were lifted for highly stable proteins (e.g. Mcm2-Mcm7) and modifications turned over by degradation (e.g. Sic1 phosphorylation). Furthermore, we mapped the effect of localisation directly on the elemental states that are responsible for localisation changes (e.g. phosphorylation of Ace2 and Swi5 in the nuclear localisation signal directly regulates their promoter recruitment; Supplementary Figure 3), bypassing the spatial description without compromising the regulatory logic. However, for most of the network we find biochemical (for reactions) or combinations of biochemical and genetic (for contingencies) data that can be formalised in the rxncon language using 14 elemental reaction types (Supplementary Data 1), creating a mechanistically detailed knowledge base of the molecular network that controls and executes the cell division cycle.

**Three independent replication programmes**. The GSM contains the current mechanistic knowledge on information transfer and, hence, connections represent direct and functional connections in vivo. Conversely, a lack of connection implies that no (known) direct and/or functional connection exists. In particular, there is very little interaction between the three duplication cycles that execute DNA replication (DNA), nuclear division (ND) comprised of SPB duplication and nuclear division itself, and cell division (CD) involving budding, morphology and cytokinesis outside of mitosis. While the lack of direct mechanistic connections between the cycles may reflect missing knowledge, the GSM predicts that these cycles constitute distinct programmes that can be executed independently through the appropriate adaptation of (the state of) the regulatory network. Such uncoupling has been observed in eukaryotic cells, leading to ploidy shifts, multi-nucleate cells, or cells without nuclei. The perhaps most prominent example of the modularity of these processes is meiosis, where a single DNA replication (pre-meiotic S-phase) is followed by two rounds of nuclear division (meiosis I + II) without any cell division[57]. Hence, the CDC consists of three independent replication cycles; DNA replication, nuclear division and cell division.

**Control points instead of a cycle**. We made a number of striking observations in the model building process. While we only used previously published data, the assembly into a holistic picture of the assembled knowledge highlighted features that were not obvious from the modules themselves. In particular, the regulatory network controlling the cell division cycle does not in itself form a cycle. Instead, it falls apart into three regulatory

modules, corresponding to the $G_1/S$, $G_2/M$ and $M/G_1$ transitions (we found no evidence for a regulatory $S/G_2$ boundary). Judged on the available data, these three subnetworks only interact indirectly through the progression of the DNA, ND and CD cycles (Fig. 2d). Arguably, this makes perfect sense, as the control network must be responsive to the state of the cell division cycle. At the same time, this is not widely recognised in the literature, and several current models of the cell division cycle directly link the three regulatory modules into a clock-like network. Our findings here suggest that this is incorrect.

**A role for the DNA damage response in normal CDC control**. The GSM connects knowledge from several distinct areas of research. When we combined knowledge from the cell cycle and DNA damage research fields, a clear picture emerged of how DNA replication is monitored and how this prevents mitotic entry. Ongoing DNA replication leads to ssDNA/Rad53 signalling from replication forks (Fig. 3). This signal maintains S-phase transcription (through inhibition of the Nrm1 repressor; Supplementary Figure 5), and inhibits the $G_2/M$ transition through transcriptional inhibition of the *CLB2* cluster (through inhibition of the Ndd1 activator; Supplementary Figure 7) and through post-translational inhibition of Cdc28-Clb2 through stabilisation of the inhibitory CDK tyrosine kinase Swe1 (Supplementary Figure 9). While the S-phase checkpoint is well known from studies on DNA damage and repair, its function is again not (widely) recognised in the literature on the cell division cycle control. However, the essential function of Mec1/Rad53 signalling in normal cell-cycle control suggests that this may be its primary function.

**Ordered CDC progression and arrest**. Next, we analysed the GSM through parameter-free simulation. The rxncon network is not directly executable, but compilable into a uniquely defined bBM[21]. We used the bBM to evaluate the completeness of the cell division cycle model in three steps: first, we searched for a point attractor in the absence of nutrients, reflecting $G_0$ arrest. Second, starting from this $G_0$ attractor, we released the cell-cycle arrest through addition of nutrients and searched for a cyclic attractor with ordered progression through the three macroscopic cycles. Third, we interrupted the cyclic attractor by adding compounds known to halt cell cycle, i.e. pheromones, HU, LatA or NOC. The first step was to find an appropriate initial state vector for the model. The model has 2378 nodes and hence $2^{2378}$ ($\sim 10^{716}$) possible state vectors, precluding an exhaustive search. Instead, we used the bBM default initial vector[21], in which all components are present in their native (unbound/unmodified) form while all modifications and bonds are absent, and all reactions are off. Interestingly, the resulting point attractor is consistent with

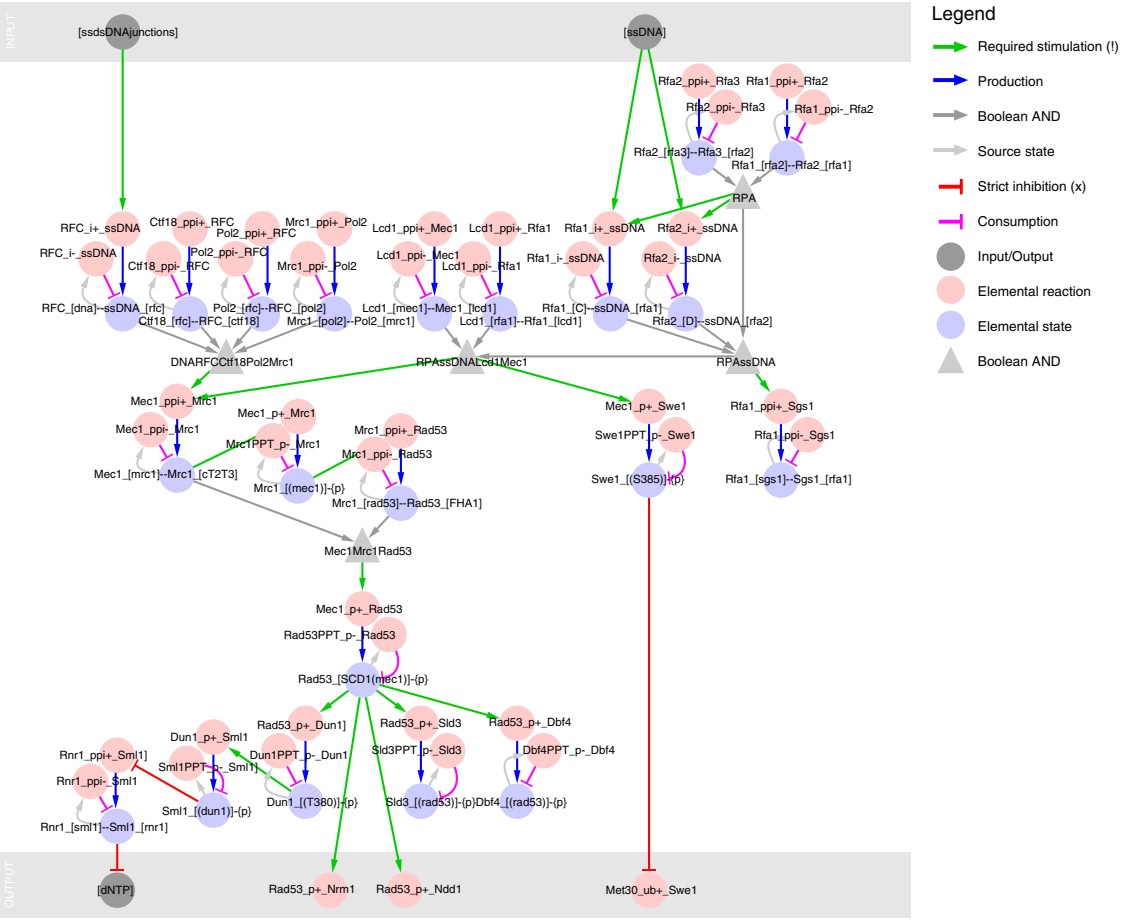

**Fig. 3** The Mec1/Rad53 pathway senses ongoing DNA replication to inhibit mitotic entry. Ongoing DNA replication leads to exposed ssDNA and ssDNA/ dsDNA junctions at the replication forks, which are sensed by the Mec1/Rad53 pathway, activating dNTP synthesis, stabilising S-phase gene expression through inhibition of Nrm1, and preventing mitotic entry through inhibition of Ndd1 and stabilisation of Swe1. The presence of ssDNA and ssDNA/dsDNA junctions allows the assembly of the ssDNA-RPA-Lcd1-Mec1 (right) and DNA-RFC-Ctf18-Pol2-Mrc1 (left) complexes, respectively. The complexes are defined through (nested) Boolean AND gates of six and four bonds, respectively. The two complexes are thought to bring active Mec1 in proximity of Mrc1, promoting their interaction and phosphorylation of Mrc1, leading to the recruitment, phosphorylation and activation of Rad53. Active Rad53 supports DNA synthesis (through Dun1 and Sml1 mediated derepression of Rnr1) and S-phase transcription (through inhibition of Nrm1, an inhibitor of the MBF transcription factor), and prevents mitotic entry (through inhibition of Ndd1, which is needed for *CLB1/2* transcription). In parallel, Mec1 prevents mitotic entry through stabilisation of Swe1, a Cdk1-Clb1/2 inhibitor. The regulatory graph is a bipartite representation where elemental reactions (red) produce or consume elemental states (blue), and elemental states influence elemental reactions through contingencies. Complex contingencies are collected in Boolean nodes (AND/OR/NOT; light grey), and input/output nodes (dark grey) define the interface between the model and its surroundings—and between the MRN and CGM parts. The figure has been simplified by omitting domain and residue names in the reaction node strings

nutrient-dependent $G_0$ arrest (Fig. 4a). Next, we simulated the network from this initial state in the presence of nutrients, and evaluated the simulation trajectory and attractor to identify inconsistencies and gaps in the knowledge base. Most importantly, the crude time concept in the Boolean model caused significant problems, as shorter event chains are faster than longer ones even if they occur through slower reactions. To resolve this problem, we introduced a timescale separation that made all transcriptional reactions and DNA replication slower than other reactions (such as post-translational modifications or protein–protein interactions), by requiring their input to be true for 20 consecutive time steps before firing (reflecting the longest signalling path). After introducing the timescale separation, which added 780 nodes to the bBM, the model accurately reflected the ordered progression through the three macroscopic cycles (Fig. 4b), resulting in a cyclic attractor with period of 186 steps. Finally, we examined the arrest points in response to pheromone, HU, LatA and NOC (Fig. 4c), identifying and

correcting one final issue in the network: the ability of Swe1 to inhibit mitotic entry in the absence of a proper bud. The changes made in the network validation phase are summarised in Table 1.

**Prediction of mutant phenotypes.** To examine the predictive power of the model, we analysed the predicted arrest point of *cdc* mutants with known phenotypes. We chose 85 (combinations of) deletions, point and constitutive (over) expression mutants, and examined their cell-cycle progression (cyclic attractor) or arrest (point attractor) (Supplementary Table 2)[19,43,58–72]. We observe two types of cyclic attractors; a normal and ordered progression through all three macroscopic cycles (viability) and a partial cyclic attractor passing through DNA replication and nuclear division, but not cell division (resulting in multinucleate cells, scored here as lethality). In addition, we observed twelve distinct point attractors (at the level of macroscopic states), which correspond to $G_1$, S, $G_2$/M, M and T arrest (Fig. 5). The model correctly

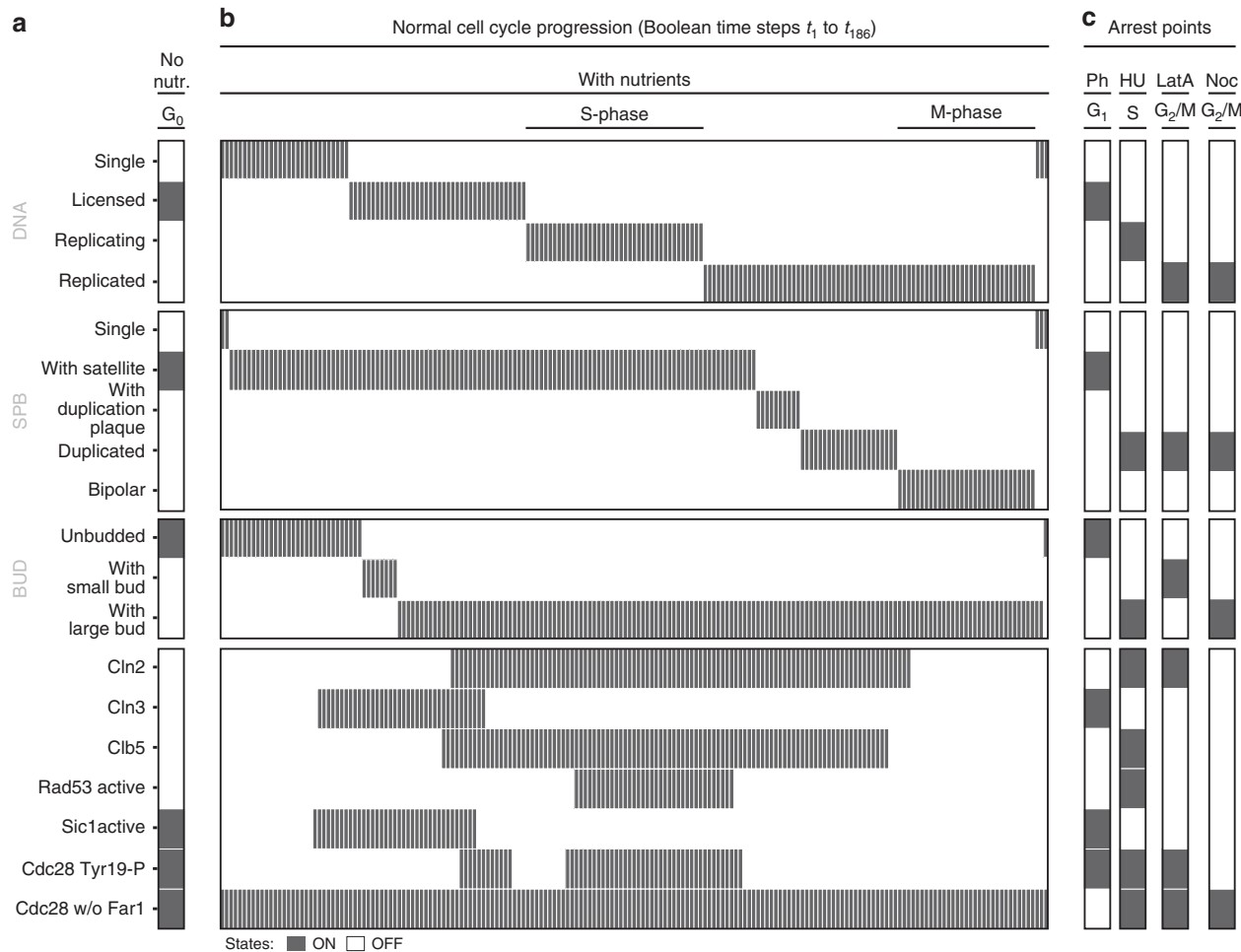

**Fig. 4** The parameter-free model reproduces both cell-cycle progression and arrest. Simulation results visualised on the CGM states (Fig. 2c), the cyclins Cln3, Cln2 and Clb5, the active form of the CDKI Sic1, and three selected elemental states: active Rad53 (indicative of ongoing DNA replication), Tyr19 phosphorylated Cdc28 (inhibition of Cdc28–Clb1/2) and Cdc28 not bound to Far1 (disappears during pheromone arrest). **a** When the model is simulated with nutrients set to false, the cells arrest at $G_0$: with licensed DNA, SPB-satellite, without bud and without cyclin or Rad53 activity. **b** When cells are released from the $G_0$ arrest by setting nutrients to True, they progress through the cell division cycle by replicating their DNA, undergoing nuclear division and rapidly thereafter cell division, and resetting the network for a new division, traversing 186 Boolean steps. Cyclin activities rise and fall through cyclin expression and inhibition by Sic1 and Swe1 (Tyr19 kinase). **c** The cells arrest again when exposed to pheromone ($G_1$: licensed DNA, SPB-satellite, no bud, only Cln3 present), HU (S: replicating DNA, duplicated SPB, large bud, Cdc28-Cln2, Cdc28-Clb5 and Rad53 are active), LatA ($G_2$/M: replicated DNA, duplicated SPB, small bud, only Cdc28-Cln2 active) and NOC ($G_2$/M: replicated DNA, duplicated SPB, large bud, and high Cdc28-Clb2 activity (not shown))

predicts all 43 lethal mutants, and 19 out of 42 viable mutants, but also scores 23 viable mutants as inviable. We note that the model is conservative in estimating viability, and look closer at the 23 inconsistent predictions (Table 2). Several of these mutants can be explained by implicit synthetic lethality, i.e. the mutant is known to be lethal in combination with a second component missing in the model. In these cases, the rescue mechanisms are not known and hence not included in the model, leading to a de facto double mutant that is correctly predicted to be lethal. For example, Clb3 and Clb4 are known to compensate for loss of Clb5, but the mechanisms are not known and, therefore, not included in the GSM. Hence, the GSM predicts the *clb5* mutant, implicitly corresponding to a *clb3clb4clb5* triple mutant, to be lethal[65]. Similarly, the mechanism by which Bck2 rescues *cln3* is unknown, explaining the lethality of the *cln3 (bck2cln3)* mutant[73]. Seven of the 23 inconsistent predictions can be explained through implicit synthetic lethality, highlighting missing mechanistic knowledge. Some of the remaining mutants can be explained by

strong phenotypes, which could be justified as lethal on a binary scale, such as the SBF-mutants (*swi4*, *swi6*, *swi6S4A* and *swi4swi6*) and *pds1*. Others, like *mbp1* and *cdh1*, which are predicted to be lethal but have no or weak reported phenotypes, likely indicate missing redundancy—again pointing to missing empirical knowledge. Taken together, the biological knowledge base uniquely defines a parameter-free model, which accurately predicts the vast majority of the mutant phenotypes that we tested—including point mutants.

## Discussion

We present a genome-scale mechanistic model of the cell division cycle. It is mechanistic, as the connections within the model correspond to direct biochemical reactions or dependencies of these reactions on the state of the reactants, down to the level of specific residues and domains. It is genome-scale, as it accounts for all components in the cell division cycle for which we could assign a mechanistic function. We visualise the complete

**Table 1 List of gap-filling additions**

| Observation | Solution | References |
|---|---|---|
| MBF target genes constitutively expressed | Swi6 phosphorylation strict requirement for MBF activity | Palumbo et al. (2016)[81] |
| Fkh2, Ndd1, Mcm1 target genes constitutively expressed | Fkh2 phosphorylation strict requirement for Fkh2 activity | Pic-Taylor et al. (2004)[82] |
| Clb2 and Clb5 not degraded in G1 | Ubiqitylation by both APC-Cdc20 and APC-Cdh1 | Wäsch and Cross (2002)[83], Lu et al. (2014)[84] |
| Pcl9 degradation not regulated | Regulated degradation equivalent to Pcl1,2 degradation | Hernandez-Ortega et al. (2013)[85] |
| Pho85-Pcl9 phosphorylation of Whi5 dominates Cdc28-mediated Whi5 phosphorylation | Pho85-Pcl9 phosphorylation positive influence on Whi5 dissociation | Hypothesis: Alternatively, Pcl9 expression could be nutrient dependent as for Cln3 |
| Sic1 deactivation premature | Phosphorylation restricted to Cdc28-Cln1,2 | Hypothesis: No Cdc28-Cln3 phosphosylation of Sic1 |
| Bud growth-module regulation insufficient | Symmetry breaking included as output/input when criteria for polarisation are fulfilled | Placeholder for quantitative phenomenon. Cannot be described mechanstically with Boolean logic |
| Stable complexes in SPB cycle disappeared | [Daughter SPB] present with duplicated and/or separated SPBs | Technical correction |
| Periodic Rad53 transcription and constant degradation lead to Rad53 periodic depletion | Constant Rad53 expression: MBF regulation as positive influence instead of strict requirement | Hypothesis: Basal Rad53 expression constitutive to account for constant role in DNA damage response |
| Discrepancy between Boolean and physiological time | Introduction of delays on transcriptional reactions | Timescale separation. Solution to technical issue with Boolean networks. |
| Dbf2 release premature | Correct spindle positioning blocks phosphorylation of Kin4 by Elm1 | Consistent with SPOC activation of phosphorylation; Caydasi et al. (2010)[86] |
| No SPB duplication due to premature Spc42 disappearance. MBF transcription very brief. | Introduction of delays on replication | Timescale separation. Solution to technical issue with Boolean networks. |
| SPB positioning premature | Kar9 phosphorylation by Cdc28–Clb1,2,5,6 | Hypothesis |
| No (morphological) cell division cycle due to premature activation of APC (due to Mad2 depletion). | Degradation of Cdc20 block when bound to Mad2 | Hypothesis |
| No (morphological) cell division cycle: cytokinesis doesn't trigger as symmetry breaking and Chs2 release never overlap | Acm1 phosphorylation completely blocks (x) instead of decreases (k-) Acm1 degradation | Hypothesis |

The table summarises the gap-filling changes made to the network in order to fully connect the regulatory logic and to reproduce wild-type behaviour: G$_0$ arrest in the absence of nutrients, ordered progression through all three macroscopic cycles in the CGM in the presence of nutrients, and proper cell-cycle arrest in response to pheromone, HU, LatA and NOC. Some of these changes are supported by non-conclusive literature evidence, some are technical modifications only included to allow meaningful bBM simulation (e.g. the time delays), and some constitute testable hypotheses. Specific additions and changes are indicated in Supplementary Data 1

knowledge base as a single connected network, and show that it defines an executable model. We use this model for validation of the network and to predict genotype-to-phenotype relationships down to the resolution of specific residues.

The genome-scale scope allows us to interpret missing features. The most striking observation is the lack of a single cycle: the control network falls apart into three distinct control circuits; G$_1$/S, G$_2$/M and M/G$_1$, which monitor and control three distinct replication cycles: DNA replication, nuclear division and cell division. While missing connections may reflect missing knowledge, the modularity we observe seems to make perfect sense: the control network needs to be more than a sizer/timer; it must respond to the replication cycles it controls. Similarly, the replication cycles must be independent to explain ploidy shifts, multinucleate (or nuclei-free) cells and meiosis. Nevertheless, the regulatory mechanisms that uncouple these cycles remain largely unknown even in baker's yeast.

These findings have implications for our understanding of the CDC. The regulatory network has previously been modelled as a closed cycle, without being explicitly dependent on the progression of the replication events. For example, it has been difficult to find a mechanistic link between the G$_1$/S (Cln1/2, Clb5/6 expression) and G$_2$/M (Clb1/2 expression) transitions. We and others have modelled this as a gene expression cascade (Clb5/6 ->Clb3/4->Clb1/2), but without being able to explain how this cascade responds to, e.g. HU-induced S-phase arrest ([74] and own

unpublished results). By combining knowledge from two fields, we explain this through Mec1/Rad53 signalling from replication forks to both transcriptional and post-translational inhibition of the Cdc28–Clb1/2 kinase complex. This knowledge was available in the literature but was only brought together to explain cell-cycle regulation through the genome-scale perspective.

The genome-scale perspective was made possible through a new approach to model construction. First, we find it indispensable to work in a text-based format, rather than model code or graphical formats, as it (i) makes the model construction, annotation, and merging process more efficient and (ii) enables processing into both graphical and executable formats. Second, it is essential that the model is built at the same resolution as the empirical data, to ensure both precision and composability—i.e., that the model entries faithfully mirror the underlying empirical data, and that they are independent of the model scope and remain unchanged as the model expands. Neither of these constraints are fulfilled in the previous (microstate) models and maps of the cell division cycle, as microstates are scope-dependent (they depend on which elemental states are included in the model) and rarely fully defined by empirical data (especially in larger models with more elemental states per component). This leads to a mix of data and assumptions in the contingency layer of these networks, which makes it very challenging to extract the actual knowledge base. Consequently, the model we present here was built independently of these previous efforts. However, this model can

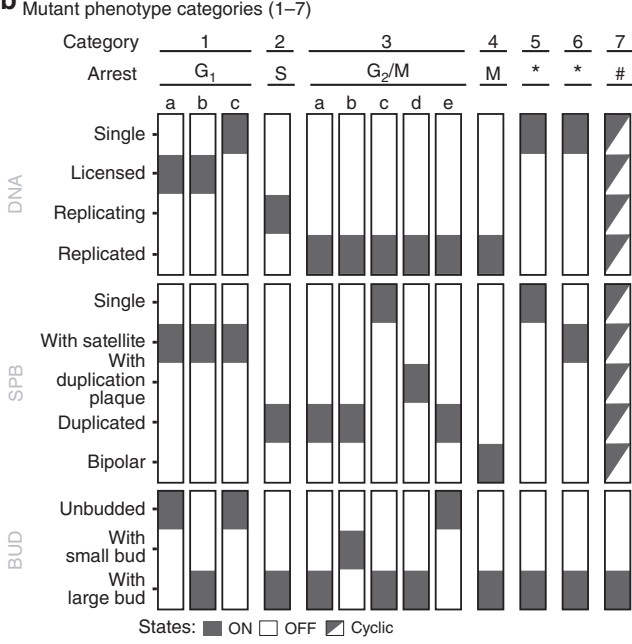

**Fig. 5** The model predicts the phenotypes of a majority of all tested mutants. Simulations were performed by introducing deletions (setting all protein states (and mRNA and gene, if appropriate) to false), point mutations (by locking proteins in a certain modification state) or constitutive overexpression (by setting the transcription reaction to True constantly), and the mutants were analysed through simulation by release from the (modified) $G_0$ state of the wild-type. **a** The model predicts the correct phenotype (lethal or viable) for 62 out of 85 mutants, although closer inspection revealed that several of the 23 inconsistent mutants can be explained by implicit synthetic lethality or strong phenotypes (Table 2). **b** The 62 mutants that are predicted to be dead arrest at thirteen distinct arrest points, that correspond to $G_1$ (1a–c, 5*, 6*), S (2), $G_2$/M (3a–e), M (4) and T (5*, 6*) arrest, or progress through a partial cyclic attractor with repeated DNA replication and nuclear division but no cell division, resulting in multinucleate cells. Note that group 5 and 6 contain both $G_1$ and T arrested cells. Both have completed DNA segregation and nuclear division, but T arrested cells fail to undergo cytokinesis and hence arrest with two nuclei. The cells in groups 5 and 6 can only be distinguished on the DNA/ nuclei count (Supplementary Table 2)

easily be maintained, modified or extended, due to the composability and careful annotation of individual elemental reactions and contingencies. These features, together with compilability into a modelling formalism that enable system level predictions directly from a qualitative description of the molecular biology, make genome-scale modelling of signal transduction possible.

Despite its scope, the GSM we present here comes with several limitations. First, it is a biological knowledge base focussed on qualitative information. Second, this information is limited by literature data coverage and quality. Although the CDC of *S. cerevisiae* is exceptionally well known, we repeatedly had to introduce gap-filling assumptions when knowledge is uncertain or missing. Third, the mutant analysis indicates missing redundancies and that certain effects cannot be properly described at the qualitative level only. Fourth, the model does not explicitly

include spatial aspects, as complex level properties (such as localisation) cannot currently be expressed in the rxncon language. Fifth and finally, the Boolean modelling logic is a very crude approximation both quantitatively and temporally, and we needed to introduce timescale separation between transcription and the other signalling events. Quantitative models will be required to analyse processes such as dynamics of cell structures or polarity establishment. However, it is currently not possible to build a quantitative model at this scale, as it would require thousands of undefined parameters values. As for metabolic networks[2], the qualitative information is more abundant and less uncertain, and qualitative models can capture the vast majority of the mutant phenotypes. Interestingly, the discrepancies between model predictions and known phenotypes primarily identified missing redundancies, i.e. missing empirical knowledge, rather than methodological limitations. Hence, this first GSM draft highlights the need for further dedicated work on the experimental characterisation of this fundamental model system.

This work is complementary to quantitative modelling efforts. While we can read out 1403 variables—elemental reactions and states—their values are limited to true or false. In contrast, quantitative models can precisely predict quantitative or complex phenotypes, such as cell-cycle duration, cell size distribution and polarity establishment[17,19,75]. However, the presented model is much larger in scope than these quantitative models, and only the Kaizu map is similar in scope although not executable[20]. One of the major outstanding challenges will be to reconcile the two: genome-scale scope and quantitative modelling. While our model could in principle be converted into an RBM[56], a number of hurdles remain: first, there is no clear equivalence to rxncon inputs in RBMs and the connections between modules would need to be hand-crafted. Second, we would need to find or estimate values for >790 unique parameters. Third, we would need representations of the DNA replication, nuclear division, and cell division cycles that are compatible with agent-based simulation. In the foreseeable future, meaningful simulation of signal transduction networks at this scale will most likely remain qualitative or semi-quantitative, as it does for metabolic networks.

Taken together, we show that it is possible to build, visualise and simulate mechanistic genome-scale models of eukaryotic signal transduction networks. Until today, this has only been done for metabolic networks, and the mass-transfer logic leads to crippling scalability issues when applied to signal transduction networks (due to microstate enumeration; reviewed in refs. [5,6,12]). We solve this technical issue by using a formalism with adaptive resolution. We chose rxncon, as it is text-based rather than graphical, uses a higher-level biological language rather than model code, supports scalable visualisation, and is compilable into a parameter-free model for direct simulation[21,76]. In particular, we find the iterative visualisation in the regulatory graph invaluable in the model construction process, and present the final model in a wall-chart that is inspired by the biochemical pathway maps[77]. In addition, the ability to go from a pure biological knowledge base to a parameter-free model that can predict system level features from molecular mechanisms is highly non-trivial[5]. Here, we use this feature for network validation—by iteratively analysing and improving the network until the corresponding model reproduces wild-type behaviour—and for genotype-to-phenotype predictions of 85 mutants. Hence, we bring together the mechanistic precision in the molecular biology, the scope of the comprehensive maps, and the executability of the mathematical models. A similar high-resolution knowledge base on the information processing network in human cells would be an important step towards a human whole-cell model and—when it accounts for the molecular effect of allele differences and drug perturbations—truly personalised medicine.

**Table 2 List of mutants with phenotypic predictions that are inconsistent with observed phenotypes**

| Mutant | Arrest point | DNA | Nuclei | Comment | Synthetic lethality |
|---|---|---|---|---|---|
| cln3 | lic, sat, 0 | 1 | 1 | G1-arrest | cln3bck2 |
| cln3mbp1 | lic, sat, 0 | 1 | 1 | G1-arrest | cln3bck2 |
| cln3swi6 | lic, sat, 0 | 1 | 1 | G1-arrest | cln3bck2 |
| cln3swi4 | lic, sat, 0 | 1 | 1 | G1-arrest | cln3bck2 |
| swi4 | lic, sat, large | 1 | 1 | G1-arrest | |
| swi6 | lic, sat, large | 1 | 1 | G1-arrest | |
| swi64A | lic, sat, large | 1 | 1 | G1-arrest | |
| mbp1 | lic, sat, large | 1 | 1 | G1-arrest | |
| clb5 | lic, sat, large | 1 | 1 | G1-arrest | clb3clb4clb5 |
| cdc20pds1clb5 | lic, sat, large | 1 | 1 | G1-arrest | clb3clb4clb5 |
| cdc20pds1clb5cdh1SIC1 | lic, sat, large | 1 | 1 | G1-arrest | clb3clb4clb5 |
| spc1102ACDK | lic, sat, large | 1 | 1 | G1-arrest | |
| Sld2A | lic, sat, large | 1 | 1 | G1-arrest | |
| Sld3T600A | lic, sat, large | 1 | 1 | G1-arrest | |
| Sld3S622A | lic, sat, large | 1 | 1 | G1-arrest | |
| Sic17A | lic, sat, large | 1 | 1 | G1-arrest | |
| cdh1 | 0, sat, 0 | 1 | 1 | G1-arrest | |
| cln1cln2cdh1 | 0, sat, 0 | 1 | 1 | G1-arrest | |
| spc1104A | rep'd, dup, large | 1 | 1 | G2/M-arrest | |
| pds1 | 0, 0, large | 2 | 2 | T arrest | |
| net1cdc15cdh1 | 0, sat, large | 2 | 2 | T arrest | |
| cdc15net1 | ?,?, large | * | * | Multinucleate: DNA replication and ND, no cell division | |
| tem1net1 | ?,?, large | * | * | Multinucleate: DNA replication and ND, no cell division | |

The model predicted 23 viable mutants to be lethal. Of these, seven mutants can be explained by implicit synthetic lethality: while cln3 and clb5 are viable, the model lacks implementation of mechanistic functions of Bck2 and Clb3/Clb4, which are essential in the absence of Cln3 and Clb5, respectively. While other mutants have strong phenotypes in, e.g. cell size or growth rate, such as swi4, swi6 and pds1, others—like cdh1—have no strong phenotypes as single mutations, suggesting that redundancy mechanisms are missing in our current knowledge

## Methods

**Model creation**. The cell division cycle model was created using the second generation rxncon language[76], using an iterative workflow described in detail elsewhere[56]. The rxncon model consists of two types of statements that both correspond directly to empirical data. First, elemental reactions define decontextualised reaction events in terms of changes in elemental states. An elemental reaction is essentially a reaction centre in a RBM[78]. Second, contingencies define constraints on the elemental reactions in terms of (Boolean combinations of) elemental states.

Elemental reactions can be either mono- or bimolecular, but always defined through two components; A and B, which are the same in monomolecular reactions. Each component is defined at a certain resolution, depending on the reaction type and the component's role in that reaction (see ref. [76] for details). Elemental reactions contain no contextual information beyond the state that changes: e.g. a phosphorylation reaction requires that the target residue is unphosphorylated. Any additional context is defined as contingencies.

Contingencies define which states (or inputs) must be true or false for an elemental reaction to take place. The rxncon language only considers direct mechanistic effects, i.e. the states must belong either to the reactants or to a complex the reactants are part of. In the latter case, complexes are defined through structured Boolean contingencies (see ref. [76] for details).

The complete set of contingencies for a single elemental reaction defines the reaction context(s) in a RBM, but in rxncon the context definition is separated over several contingencies that each define the impact of a single elemental state[76]. This gives a one-to-one correspondence between model definition and empirical data, improving composability as model changes and extension have a local impact only. Multiple contingencies can be combined into statements that are arbitrarily complex, defining reactions down to microstate resolution when necessary. Typically, however, single or few contingencies per elemental reaction suffice to capture all the empirically known regulatory mechanisms. In the end, the rxncon model constitutes a molecular biology knowledge base in formal language, which is computer readable, can be automatically visualised and compiled into a uniquely defined executable model.

For a detailed description of how to build a rxncon model, see ref. [56].

**Model simulation**. Parameter-free models were created automatically using the rxncon compiler[21] (Supplementary Data 3–5), and simulated with the R package BoolNet[79]. The rxncon software can be downloaded from https://github.com/rxncon/rxncon or directly installed from the python package index ("pip install rxncon"). See https://rxncon.org for further instructions.

The initial $G_0$ simulation was performed with the default initial vector (Supplementary Data 4), except that the placeholder input [Histones] was set to true. This simulation resulted in a point attractor considered to correspond to $G_0$ arrest in the absence of nutrients. The following simulations were performed from this starting state with nutrients set to true only (for normal cell-cycle progression), or in combination with either chemical inputs (Pheromone, HU, LatA or NOC) set to true or with changes reflecting the mutations (all states of deleted components

and their genes/mRNA when applicable) set to false; phosphorylated/unphosphorylated states set to constant true/false for alanine or phosphomimetic residue changes; transcription set to constant true for (constant) over expression. The final attractor and path to attractor were analysed for each condition to determine phenotypes and arrest point.

**Network visualisation**. The cell division cycle model was visualised as rxncon regulatory graphs[10], using Cytoscape (http://www.cytoscape.org/) and a visual formatting file (rxncon2cytoscape.xml; https://github.com/rxncon/tools).

**Model file and references**. The cell division cycle model is compiled in an SBtab compatible spreadsheet[80], and is available in Supplementary Data 1 and from https://github.com/rxncon/models (CDC_S_cerevisiae.xls). The model is fully referenced through the reference columns in the reaction and contingency sheet, and explained in detail in the Supplementary Methods.

The model is based on the literature listed in Supplementary Table 1, as specified for individual reactions and contingencies in Supplementary Data 1.

**Reporting Summary**. Further information on experimental design is available in the Nature Research Reporting Summary linked to this article.

**Code availability**. The models and code are freely available through the paper or public repositories. The rxncon software is open source, distributed under the lGPL licence, and can either be downloaded from https://github.com/rxncon/rxncon or installed from the python package index with "pip install rxncon". The rxncon model file is available as Supplementary Data 1 or through download from https://github.com/rxncon/models/.

## Data availability

No new data were generated in this study. The model is available through the paper or public repositories. The rxncon model file is available as Supplementary Data 1 or through download from https://github.com/rxncon/models/.

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

## Acknowledgements
We would like to thank Jesper Romers, Sebastian Thieme and Mathias Wajnberg for close collaboration in the methods development, and Matteo Barberis for critical comments on the network model. M.K. would like to thank Hiroaki Kitano and Stefan Hohmann for the inspiration and support to tackle the challenge of large-scale signalling networks. This work was supported by the German Federal Ministry of Education and Research via e:Bio Cellemental (FKZ0316193, to M.K.).

## Author contributions
U.M. and M.K. built and analysed the model, with input from E.K. M.K. drafted the manuscript with input from all authors. All authors read and approved the final manuscript.

## Additional information

**Competing interests:** The authors declare no competing interests.

