## [Peer Review File · Nature Communications]

Reviewers' comments:

Reviewer #1 (Remarks to the Author):

The submission of Münzner et al. presents a large scale reconstruction of the *Saccharomyces cerevisiae* (bakers' yeast) cell cycle that brings together a large body of knowledge concerning the biochemical and cell biological components of this widely-studied and fundamental biological process. The resulting model contains 790 reactions that govern the activity of 1238 model components (representing proteins, protein modification sites, mRNA as well as more coarse-grained properties of the cell such as DNA state, budding state, etc.). The description of processes in the form of reactions is augmented by so-called contingencies that allow the effects of multiple component states to affect each reaction, thus allowing for a detailed description of the molecular biochemistry, which is required to accurately represent the underlying biology and our knowledge of it. The use of such a language provides precision and rigor to the formulation of the model and also provides one of the major advantages of this reconstruction over previous efforts, which is that the model can be directly simulated and used to predict the phenotypic behavior of the cell under different environmental and genetic conditions. The manuscript presents such simulations, which are used to test the model against the known phenotypes of 85 different yeast mutant strains. Although the simulation approach used for the model is highly coarse-grained — each model variable can only take the two values TRUE or FALSE — the model correctly represents the known phenotype in 62/85 (72%) tested strains. Moreover, the simulation approach leads to identification of gaps or inaccuracies in the initial reconstruction. The manuscript contains an extensive supplement that fully documents the model, which is itself provided as a spreadsheet that can be used as input to a model simulator that is also freely available and open-source and which has been described previously by the authors. The model thus represents a valuable resource to the biological community and its presentation and description should be of general interest to readers. The paper is generally clear and well-written for a general audience. That said, I feel that the presentation of the model should be improved to make it more understandable to an audience that is not already well-versed in the rxncon formalism. I give some detailed suggestions on that below as well as raise some other points that I think the authors should address.

Major

1. It's very difficult if not impossible for a general reader to understand the model based on what is presented in the main figures on the text. Figure 1a is only useful insofar as it conveys the overall modular architecture of the model superimposed on a much more detailed reaction network. Figure S1 provides a high resolution version of this figure. These cannot be understood, however, without a basic understanding of the rxncon language and its visual representation in the form of the regulatory graph. This language, though useful, is not widely known among biologists. Therefore, some introduction is necessary before expecting the reader to get much out of these diagrams. Figure 2 does provide some explanation of the visual representation, but suffers from several problems. First, the meaning of the text labels on the nodes is not explained in the captions or the text. Instead the reader is referred to a preprint (or a chapter from an as yet unpublished book). I don't have a problem with the publication format of these materials, just with the fact that the paper as it stands is not reasonably self-contained. Second, to add to a reader's potential confusion, there is an error in the legend that mixes up reaction and state. Third, the labeling of the graph is very cluttered, there seems to be a lot of redundancy between the labels and the nodes they are representing (e.g., to the elemental states of the reactants need to be included in the node labels for the reactions in every case?). I think most readers will find Figure 2 very challenging to understand, even at a high level. At a minimum there need to be some sub panels introducing the formalism before it is presented as content that a reader is expected to get something from, i.e. before Figure 1b. One suggestion would be to switch the order of Figures 1 and 2 and their corresponding subsections in Results.

-Figure 1c is key to a central point of the paper — that the checkpoints of the cell cycle are not

directly connected through regulatory processes but rather through the progression of the states of the respective processes they regulate — but I couldn't really understand what was going on in this figure or how its nodes and processes related to the reconstructed model. The meaning of the different node types and their relationships as well as correspondence to model components needs to be better explained.

-Figure 1d is currently not referenced in the text. Presumably its importance should be part of the "Building a mechanistic knowledge base at genome scale" section.

-Figure 3 is again hard to understand because of the use of rxncon notation to label the model variables being plotted. The labels are also quite small.

-Same thing applies to Figure 4b. The labels on the x-axis are so tiny that I didn't see them at first and was confused.

Figure 4a: I have no idea what is being shown except that the numbers somehow correspond to the predictions of mutant phenotypes.

2. I would like the authors to comment a bit more on the relationship between the current effort and the previous work of Kaizu et al. (ref. 20). Beyond the issue of executability, was there anything different about the approaches? How does the size and scope differ? To what extent did the current effort build on the previous one and if the efforts were largely independent, why was it not possible to build on the earlier work? I think these questions are central to the issue of genome-wide reconstruction of such regulatory networks.

Minor

-I couldn't find anywhere a reference to the mutant library first referred to at Line 80 and then presented in Tables 2 and S3.

-Lines 102-103 refer to "elemental reactions," "elemental states," and "contingencies" in reference to the model, but these terms aren't described until line 121ff.

-There are several references in the text to Supplemental Figures that are not part of the submission (not listed in the list of Figures and not provided in the files). These occur at lines 133, 176, 178, and 179. I believe that these are references to figures in the Supplemental Text description of the model. This needs to be clarified. It could help in general to have a sentence or two in the main text laying out what is in the Supplemental Text description of the model and how to use it as a reference.

-Line 196, "Instead, we used the bBM default vector²¹." I think a sentence or two explaining the logic for the default selection of initial conditions would be useful here instead of just a reference.

-Line 223. The term T-arrest is used without being defined.

-Table of contents for SI Text would be helpful.

Reviewer #2 (Remarks to the Author):

The authors present a mechanistic modeling framework to enable genome-scale models of the budding yeast cell cycle. The framework is aimed at solving scalability issues suffered by parameterized models of the cell cycle (e.g. over-fitting, parameter inference). Although reported to interrogate mechanistic relationships, the qualitative model is very coarse, and thus is unlikely to reveal new mechanistic insights into the control of the cell division cycle. At this time it is not clear to me that this modeling framework constitutes a substantial advancement over other

modeling work (e.g. John Tyson's group using more traditional ODE modeling techniques). In the end, this manuscript reads like a very sparse literature review of cell cycle control and simply describes what was already known in a network model. While these types of models undoubtedly be valuable as we begin to contend with the scales of interactions at the organismal level, this manuscript will likely be of greatest interest a modeling community, and it is not at a point where the cell cycle community will observe any benefits.

Specific points.

1. Given that the authors have presented a complex model with many interactions gleaned from literature, and that they tested the model against 85 cell cycle mutations, one would expect more than 33 references. I suspect this is a cultural issue related to how those in the mathematics field assign credit. Nonetheless, without substantial referencing, it is impossible to evaluate how the authors assigned the associations presented in their mechanistic networks.

2. The authors highlight the fact that their model fell out into 3 replication modules that could function independently and the transitions were gated by checkpoints, not directly linked. The authors suggest that this observation is not widely recognized in the literature, which is odd, as Hartwell won the Nobel Prize, in part for his discovery that cell cycle events were not always "dependent" like substrate – product relationships but were ordered by checkpoints. The authors also seem to suggest that these checkpoints are key "ordering" events. However, the authors did not address the observation that mutations in checkpoint pathways (like those enforced by Rad53) have little effect on unperturbed cell cycles. These observations indicate that checkpoint-independent ordering mechanisms are available.

Reviewers' comments:

Reviewer #1 (Remarks to the Author):

The submission of Münzner et al. presents a large scale reconstruction of the *Saccharomyces cerevisiae* (bakers' yeast) cell cycle that brings together a large body of knowledge concerning the biochemical and cell biological components of this widely-studied and fundamental biological process. The resulting model contains 790 reactions that govern the activity of 1238 model components (representing proteins, protein modification sites, mRNA as well as more coarse-grained properties of the cell such as DNA state, budding state, etc.). The description of processes in the form of reactions is augmented by so-called contingencies that allow the effects of multiple component states to affect each reaction, thus allowing for a detailed description of the molecular biochemistry, which is required to accurately represent the underlying biology and our knowledge of it. The use of such a language provides precision and rigor to the formulation of the model and also provides one of the major advantages of this reconstruction over previous efforts, which is that the model can be directly simulated and used to predict the phenotypic behavior of the cell under different environmental and genetic conditions. The manuscript presents such simulations, which are used to test the model against the known phenotypes of 85 different yeast mutant strains. Although the simulation approach used for the model is highly coarse-grained — each model variable can only take the two values TRUE or FALSE — the model correctly represents the known phenotype in 62/85 (72%) tested strains. Moreover, the simulation approach leads to identification of gaps or inaccuracies in the initial reconstruction. The manuscript contains an extensive supplement that fully documents the model, which is itself provided as a spreadsheet that can be used as input to a model simulator that is also freely available and open-source and which has been described previously by the authors. The model thus represents a valuable resource to the biological community and its presentation and description should be of general interest to readers. The paper is generally clear and well-written for a general audience. That said, I feel that the presentation of the model should be improved to make it more understandable to an audience that is not already well-versed in the rxncon formalism. I give some detailed suggestions on that below as well as raise some other points that I think the authors should address.

Answer: Thank you. We appreciate the effort the reviewer has made, as well as the constructive criticism and helpful suggestions. We have improved the paper accordingly, as described in detail below.

Major

1. It's very difficult if not impossible for a general reader to understand the model based on what is presented in the main figures on the text.

Answer: We are aware of this, but a model at this scale cannot be explained in detail within the page limits of a normal research paper. Consequently, we have focussed the paper and its figures on the

key features and principal architecture of the network and model. The detailed description of the biological knowledge and model is, as the reviewer points out, in the supplemental text and figures. However, we have improved the presentation according to the reviewer's suggestions, as detailed below.

Figure 1a is only useful insofar as it conveys the overall modular architecture of the model superimposed on a much more detailed reaction network. Figure S1 provides a high resolution version of this figure. These cannot be understood, however, without a basic understanding of the rxncon language and its visual representation in the form of the regulatory graph. This language, though useful, is not widely known among biologists. Therefore, some introduction is necessary before expecting the reader to get much out of these diagrams.

Answer: We agree with the reviewer. We introduced an additional Figure, now Figure 1, which presents the key concepts of the rxncon language and the regulatory graph. Also, the purpose of Figure 1A (now 2A) is to give a model overview and to advertise the high-resolution version in Figure S1. We added a reference to Fig S1 in the legend to make this clearer:

“Bird’s eye view of the complete network (see Fig S1 for a readable poster-sized version).”

Figure 2 does provide some explanation of the visual representation, but suffers from several problems. First, the meaning of the text labels on the nodes is not explained in the captions or the text. Instead the reader is referred to a preprint (or a chapter from an as yet unpublished book). I don't have a problem with the publication format of these materials, just with the fact that the paper as it stands is not reasonably self-contained.

Answer: As mentioned above, we added a new figure (Figure 1) that introduces the regulatory graph and the rxncon language. This introduction, although brief, should cover all the essential information needed to read the model and figures, including the meaning of the text labels.

Second, to add to a reader's potential confusion, there is an error in the legend that mixes up reaction and state.

Answer: We are impressed by the thoroughness of the reviewer. We have corrected the legend. Thank you.

Third, the labeling of the graph is very cluttered, there seems to be a lot of redundancy between the labels and the nodes they are representing (e.g., to the elemental states of the reactants need to be included in the node labels for the reactions in every case?).

Answer: We agree with the reviewer; the labelling - which carries essential information about the graph - becomes very cluttered in such dense graphs. We have now reduced the complexity by

omitting domain and residue information in the reaction labels, which is redundant with the information in the state nodes.

I think most readers will find Figure 2 very challenging to understand, even at a high level. At a minimum there need to be some sub panels introducing the formalism before it is presented as content that a reader is expected to get something from, i.e. before Figure 1b. One suggestion would be to switch the order of Figures 1 and 2 and their corresponding subsections in Results.

Answer: As mentioned above, we added a completely new Figure, Figure 1, to explain the notation. In addition, we have extended the legend to explain the signalling cascade in more detail.

-Figure 1c is key to a central point of the paper — that the checkpoints of the cell cycle are not directly connected through regulatory processes but rather through the progression of the states of the respective processes they regulate — but I couldn't really understand what was going on in this figure or how its nodes and processes related to the reconstructed model. The meaning of the different node types and their relationships as well as correspondence to model components needs to be better explained.

Answer: We have now redrawn this figure using the CGM states in panel B, indicating which transitions are directly regulated by the control network, and which of the CGM states are necessary to allow the next transition. In addition, we have extended the explanation in the legend, explicitly stating the key events at each transition point.

-Figure 1d is currently not referenced in the text. Presumably its importance should be part of the "Building a mechanistic knowledge base at genome scale" section.

Answer: We have removed the figure as the content is covered by the new Figure 1.

-Figure 3 is again hard to understand because of the use of rxncon notation to label the model variables being plotted. The labels are also quite small.

Answer: We have reworked the figure design to clarify both the variables shown (using verbal descriptions rather than model states as labels) and by adding additional descriptors at the top.

-Same thing applies to Figure 4b. The labels on the x-axis are so tiny that I didn't see them at first and was confused.

Answer: We have reworked the figure design to clarify both the variables plotted and the categories of mutant arrest, and we have moved the labels to the top.

Figure 4a: I have no idea what is being shown except that the numbers somehow correspond to the predictions of mutant phenotypes.

Answer: The figure shows the model predictions (point attractor = dead, cyclic attractor = alive) vs. the *in vivo* mutant phenotypes (viable, lethal). We have clarified this in the labelling.

2. I would like the authors to comment a bit more on the relationship between the current effort and the previous work of Kaizu et al. (ref. 20). Beyond the issue of executability, was there anything different about the approaches? How does the size and scope differ? To what extent did the current effort build on the previous one and if the efforts were largely independent, why was it not possible to build on the earlier work? I think these questions are central to the issue of genome-wide reconstruction of such regulatory networks.

Answer: We have now added a paragraph to the discussion elaborating on our experiences with the reconstruction and what we think are essential for genome-wide reconstruction of regulatory networks:

“The genome-scale perspective was made possible through a new approach to model construction. First, we find it indispensable to work in a text based format, rather than model code or graphical formats, as it (i) makes the model construction, annotation, and merging process more efficient and (ii) enables processing into both graphical and executable formats. Second, it is essential that the model is built at the same resolution as the empirical data, to ensure both precision and composability - i.e., that the model entries faithfully mirrors the underlying empirical data, and that they are independent of the model scope and remain unchanged as the model expands. Neither of these constraints are fulfilled in the previous (microstate) models and maps of the cell division cycle, as microstates are scope dependent (they depend on which elemental states are included in the model) and rarely fully defined by empirical data (especially in larger models with more elemental states per component). This leads to a mix of data and assumptions in the contingency layer of these networks, which makes it very challenging to extract the actual knowledge base. Consequently, the model we present here was built independently of previous efforts. However, this model can easily be maintained, modified or extended, due to the composability and careful annotation of individual elemental reactions and contingencies. These features, together with compilability into a modelling formalism that enable system level predictions directly from a qualitative description of the molecular biology, make genome-scale modelling of signal transduction possible.”

Direct comparison of the two models is difficult, as the Kaizu network also includes signalling pathways (accounting for ca 200 of the 449 components in the map) in addition to cell cycle regulation. Furthermore, some of the Kaizu-unique components correspond to protein complexes in our model, such as the APC monomers. Comparison at the level of reactions (732 in the Kaizu model) is also complicated due to the difference between elemental and microstate reactions.

Minor

-I couldn't find anywhere a reference to the mutant library first referred to at Line 80 and then presented in Tables 2 and S3.

Answer: We have now assigned the reference to each mutant in Table S3, and cite these papers in the main text.

-Lines 102-103 refer to "elemental reactions," "elemental states," and "contingencies" in reference to the model, but these terms aren't described until line 121ff.

Answer: These terms are now introduced in the introduction, through the new Figure 1.

-There are several references in the text to Supplemental Figures that are not part of the submission (not listed in the list of Figures and not provided in the files). These occur at lines 133, 176, 178, and 179. I believe that these are references to figures in the Supplemental Text description of the model. This needs to be clarified. It could help in general to have a sentence or two in the main text laying out what is in the Supplemental Text description of the model and how to use it as a reference.

Answer: Correct. We now explicitly state this in the first results section:

"The biology and implementation is described in detail in the extensive Supplementary Information, where the network is divided into thirty smaller modules that are described and visualised individually (Fig S2-S31; Supplementary Information)."

-Line 196, "Instead, we used the bBM default vector²¹." I think a sentence or two explaining the logic for the default selection of initial conditions would be useful here instead of just a reference.

Answer: We have now added a brief explanation, which reads:

"Instead, we used the bBM default initial vector²¹, in which all components are present in their native (unbound/unmodified) form while all modifications and bonds are absent, and all reactions are off."

-Line 223. The term T-arrest is used without being defined.

Answer: We have now elaborated on this in the legend of Figure 5:

“Note that group 5 and 6 contains both G1 and T arrested cells. Both have completed DNA segregation and nuclear division, but T arrested cells fail to undergo cytokinesis and hence arrest with two nuclei. They cells in group 5 and 6 can only be distinguished on the DNA/Nuclei count (Table S3).”

-Table of contents for SI Text would be helpful.

Answer: We have added a table of content to the SI.

Reviewer #2 (Remarks to the Author):

The authors present a mechanistic modeling framework to enable genome-scale models of the budding yeast cell cycle. The framework is aimed at solving scalability issues suffered by parameterized models of the cell cycle (e.g. over-fitting, parameter inference). Although reported to interrogate mechanistic relationships, the qualitative model is very coarse, and thus is unlikely to reveal new mechanistic insights into the control of the cell division cycle.

Answer: This is only partially true. While the Boolean simulation logic is coarse, this is not true for the model or its underlying knowledge base. We include mechanistic detail down to the function of specific modifications and bonds, at specific residues and domain when known in the literature. This is at least as detailed as the Tyson model the reviewer cites below, but at a much, much larger scale. Hence, we present a knowledge base with an unrivalled combination of mechanistic detail and coverage, which should provide a high-value resource for the community in itself.

However, we also demonstrate the value of the bipartite Boolean logic in analysing this knowledge base. First and foremost, we use it for gap-finding and gap-filling, by identifying missing connections in the known information transfer. Based on these findings, we can discriminate and reconcile inconsistent literature evidence and adapt the model with hypothetical reactions and contingencies. These must be validated or rejected through dedicated empirical analysis, but this is true regardless of modelling method. Second, we use the model to predict genotype-to-phenotype relationships down to the effect of (combinations of) point mutations, and we use this feature to further validate the knowledge base. Neither has been possible for signalling models at this scale before, and hence the work we present here constitutes a significant advance beyond the previous state of the art.

At this time it is not clear to me that this modeling framework constitutes a substantial advancement over other modeling work (e.g. John Tyson's group using more traditional ODE modeling techniques).

Answer: The modelling framework we use here has two major advantages over e.g. ODE modelling. First and foremost, it makes simulation of models at this scale possible at all. For good reasons (parametrisation; simulation cost), no ODE models have been built and analysed at this scale. Second, it can be used to predict system level behaviour directly from qualitative mechanistic data. I.e., parametrisation - which introduces a second layer of uncertainty - is not needed.

ODE modelling - or quantitative modelling of any kind - will only provide a meaningful improvement when parameter values are known from empirical data or can be reasonably constrained by fitting. We are currently very far from this scenario, as reliable parameter measurements are rare, and as parameter fitting with models at this scale constitutes a massively underdetermined problem.

In the end, this manuscript reads like a very sparse literature review of cell cycle control and simply describes what was already known in a network model. While these types of models undoubtedly be valuable as we begin to contend with the scales of interactions at the organismal level, this manuscript will likely be of greatest interest a modeling community, and it is not at a point where the cell cycle community will observe any benefits.

Answer: We find it difficult to understand this criticism. It is true that the backbone of the paper is the establishment of a highly detailed knowledge base from literature knowledge, and that this knowledge per definition has been published. However, it has never before been compiled in a single knowledge base that can be visualised and executed - three features that have proven their value over and over again in the metabolic research field since the day of the first biochemical pathways maps.

We also show that value here, by using both visualisation and executability to identify gaps in the information transfer, which we close with dedicated literature review and/or hypothetical model adaptations (that are clearly annotated as such). Also, we use the executability to analyse the network structure and find that many possible missing connections may make sense (e.g. the Clb5/6 -> Clb3/4 -> Clb1/2 chain discussed in the literature) when we consider the interaction between the control network and the three division cycles.

Finally, the network is based on an in-depth literature review of several hundred papers of which we only cite the 229 from which we extracted mechanistic data (see Tables S1 and S2, and the supplementary information). While there are many cell cycle related publications we did not manage to consider, it hardly qualifies as "very sparse" review. In addition, the underlying knowledge-base (Table S1) is a much more stringent formulation of the known molecular biology than any cell cycle review we have encountered. We are confident that the cell cycle community can put this knowledge repository to good use.

Specific points.

1. Given that the authors have presented a complex model with many interactions gleaned from literature, and that they tested the model against 85 cell cycle mutations, one would expect more than 33 references. I suspect this is a cultural issue related to how those in the mathematics field assign credit. Nonetheless, without substantial referencing, it is impossible to evaluate how the authors assigned the associations presented in their mechanistic networks.

Answer: As mentioned above, we cite 229 reference in the model (Table S1 and S2) and 177 in the detailed model description (supplementary information), and went through many, many more from which we could not extract information on molecular mechanisms (and hence do not cite). The 33 references are only supporting the discussion in the main text, not the model itself. Importantly, every single model entry is individually referenced (see the reference column in the reaction and contingency lists in Table S1). It would not be possible to reference the model more stringently.

We agree with the reviewer that it would be appropriate to acknowledge these papers in the main manuscript too, and would happily reference them in the results section - however this would bring us to ca 250 references, as compared to the normal journal max of 70.

2. The authors highlight the fact that their model fell out into 3 replication modules that could function independently and the transitions were gated by checkpoints, not directly linked. The authors suggest that this observation is not widely recognized in the literature, which is odd, as Hartwell won the Nobel Prize, in part for his discovery that cell cycle events were not always “dependent” like substrate – product relationships but were ordered by checkpoints.

Answer: The reviewer is of course correct that the notion of checkpoints is well established. The point we are making is that the control network - which is widely seen as a sizer-timer “clock” that can be arrested at certain places if the appropriate checkpoints are triggered - isn’t a closed circuit. What we show here is that there appear to be no single cell cycle control network that progresses in the absence of arrest signals, but instead a set of disjunct regulatory modules or pathways that communicate through three principally independent replication cycles. This notion is definitely not well established in the literature. We have replaced our use of “checkpoint” with “regulatory modules” to make this distinction clear.

The authors also seem to suggest that these checkpoints are key “ordering” events. However, the authors did not address the observation that mutations in checkpoint pathways (like those enforced by Rad53) have little effect on unperturbed cell cycles. These observations indicate that checkpoint-independent ordering mechanisms are available.

Answer: This is correct, checkpoints - in the classical sense - seem to constitute key ordering events. The Rad53 pathway is an excellent example, as it communicates the state of the DNA to maintain S-phase and prevent Mitotic entry. This pathway is always essential, as *mec1* and *rad53* mutants are lethal. This is due to an essential function in dNTP synthesis (through inhibition of the RNR inhibitor Sml1) that can be rescued by *SML1* deletion. Such a double mutant is viable, unless DNA replication is inhibited or delayed through e.g. hydroxyurea (HU) treatment. The question of the reviewer is how we can explain this.

Most likely, this is due to the natural timing, as there is a natural delay between S-phase entry and mitotic entry (due to requirement for SPB duplication, bud growth and the sequential activation of the transcription factors needed to trigger the mitotic entry). If this natural delay is sufficient to allow DNA replication in cells with uncontrolled dNTP synthesis (*sm1* mutants), then the absence of the checkpoint should not cause a phenotype. Only if DNA replication was delayed long enough for all other requirements to be fulfilled, e.g. through HU treatment, would enough cells fail at mitosis to show a detectable phenotype.

REVIEWERS' COMMENTS:

Reviewer #1 (Remarks to the Author):

The authors have done an excellent job at addressing the points raised in my review. I feel that the manuscript is significantly improved in its accessibility for a general audience, and it provides both a useful resource for the yeast cell cycling community and a template for efforts to compile such resources to other cell regulatory and signaling systems. I also thought that the authors did a reasonable job in replying to the points raised by Reviewer 2, who expressed a number of concerns about the significance of the effort. Reviewer 2 did make one specific point about references (point 1 under Specific Points), that perhaps the editors could assist in resolving. The issue is that although around 200 references are cited in the detailed model description and Supplementary Tables that define the model, only 33 references are given in the main text, and the journal imposes an upper limit of 70 or so references for normal articles. Given the resource nature of the current work, I wonder if this limit could be extended to allow the authors to properly cite the knowledge base upon which they built the current model. In addition to allowing the authors to give proper credit to the authors of these works, the expanded reference list would leave a citation pattern in the literature that would connect mechanisms and model resources. I'm thinking it would be useful if when someone checks the citations to the fundamental papers on cell cycle mechanisms that they would see that the current paper and model cite that paper, which could lead to a higher level of interest and utilization of the resource.

Reviewer #2 (Remarks to the Author):

The author's response to the original critiques certainly addressed the concern regarding the lack of references in the main text. It does seem odd that references to fundamental cell cycle models are relegated to the supplement, but I do understand the focus needed to fit the information within the page limits. That said, this is the only concern that was satisfactorily addressed in the response.

The most fundamental criticism in the initial review was that the findings presented were more of a cell cycle review and an attempt to validate the model by fitting the phenotypes (crudely into 2 bins: live or dead) a variety of cell cycle mutants. The "novel finding" suggested by the authors is that there are separable cycles (DNA replication, nuclear division and, and cell division). As indicated in the original review, the field has known about separable modules for decades. Several conditions have been reported in which endocycling occurs suggesting DNA replication can cycle independent of mitosis and cell division. The authors describe their understanding of endocycling in the discussion of the manuscript "While missing connections may reflect missing knowledge, the modularity we observe seems to make perfect sense: The control network needs to be more than a sizer/timer; it must respond to the replication cycles it controls. Similarly, the replication cycles must be independent to explain ploidy shifts, multi-nucleate (or nuclei-free) cells and meiosis. Nevertheless, the regulatory mechanisms that uncouple these cycles remain largely unknown even in baker's yeast." The first two statements are true, but the last is inaccurate. The role/mechanism of B-cyclins in inhibiting re-initiation of DNA replication is well-established, thus oscillation of these cyclins can drive endocycles, and this has phenomenon has been established clearly in the literature. Similar findings have been reported for SPB duplication and spindle assembly, both are components of the nuclear division cycle and can cycle independent from cell division.

Moreover, in 1995, the Johnston group demonstrated that some *cdc6* mutants (I believe *cdc7* mutants as well) do not replicate DNA but move forward into mitosis and undergo a mitotic catastrophe. These findings demonstrated many years ago that mitosis can be initiated independent of DNA replication. Evidence suggests that the initiation of DNA replication is required for the DNA replication checkpoint to be activated and block the mitotic cycle. So again, the molecular mechanisms that couple/uncouple these cycles are well documented and involve the

activities of CDKs and/or checkpoints.

In the original review it was pointed out that even without checkpoints, these “independent” cycles remain ordered and cells are quite viable in the absence of perturbations. In response the authors state; “Most likely, this is due to the natural timing, as there is a natural delay between S-phase entry and mitotic entry (due to requirement for SPB duplication, bud growth and the sequential activation of the transcription factors needed to trigger the mitotic entry).” This explanation suggests some “natural timing” that offsets S-phase and M-phase entry, which is exactly the prevailing model... that there is a timer/sizer mechanism in the cell cycle control network that maintains ordering of events, and when perturbations occur, checkpoints maintain that ordering. Certainly, there are several reports that cascading transcription may be an essential part of this timing mechanism.

In the end, the field has already embraced the idea of separable cycles and have identified mechanisms that keep them connected, so I don't understand what novel findings have been presented.

I understand and appreciate that the approach pushes the boundaries of modeling (especially at scale) and may have interest to the modeling field, but without some new (and validated finding) it's not clear why this work would be of interest to the yeast cell cycle field. Moreover, the volumes of data of different types used to construct the model are not available in nearly the depth or breadth in other systems (e.g. mammalian cell culture), so it is not clear that this approach will be generalizable to systems where the details of cell cycle control are more poorly understood.

REVIEWERS' COMMENTS:

Reviewer #1 (Remarks to the Author):

The authors have done an excellent job at addressing the points raised in my review. I feel that the manuscript is significantly improved in its accessibility for a general audience, and it provides both a useful resource for the yeast cell cycling community and a template for efforts to compile such resources to other cell regulatory and signaling systems. I also thought that the authors did a reasonable job in replying to the points raised by Reviewer 2, who expressed a number of concerns about the significance of the effort. Reviewer 2 did make one specific point about references (point 1 under Specific Points), that perhaps the editors could assist in resolving. The issue is that although around 200 references are cited in the detailed model description and Supplementary Tables that define the model, only 33 references are given in the main text, and the journal imposes an upper limit of 70 or so references for normal articles. Given the resource nature of the current work, I wonder if this limit could be extended to allow the authors to properly cite the knowledge base upon which they built the current model. In addition to allowing the authors to give proper credit to the authors of these works, the expanded reference list would leave a citation pattern in the literature that would connect mechanisms and model resources. I'm thinking it would be useful if when someone checks the citations to the fundamental papers on cell cycle mechanisms that they would see that the current paper and model cite that paper, which could lead to a higher level of interest and utilization of the resource.

Answer: Thank you.

Reviewer #2 (Remarks to the Author):

The author's response to the original critiques certainly addressed the concern regarding the lack of references in the main text. It does seem odd that references to fundamental cell cycle models are relegated to the supplement, but I do understand the focus needed to fit the information within the page limits. That said, this is the only concern that was satisfactorily addressed in the response.

The most fundamental criticism in the initial review was that the findings presented were more of a cell cycle review and an attempt to validate the model by fitting the phenotypes (crudely into 2 bins: live or dead) a variety of cell cycle mutants. The "novel finding" suggested by the authors is that there are separable cycles (DNA replication, nuclear division and, and cell division). As indicated in the original review, the field has known about separable modules for decades.

Answer: The novelty of the model presented in the paper is much greater than the reviewer acknowledges.

Most importantly, it is the first genome-scale mechanistic model (GSM) of any signalling system. GSMs (and smaller rxncon models) are not optimised or fitted to phenotypes. They are simply very precise formulations of molecular biology knowledge. These started with the Biochemical Pathway posters, and has until this work only been created for metabolic networks - but they have had an immense impact on the metabolic research field. Here, we show that - given an appropriate formalism - GSMs can be constructed also for signalling networks.

The comparison of a GSM to a review is also a far stretch. While the GSM, like a review, builds on existing knowledge, the GSM has a much, much higher requirement for specificity and precision. In the GSM, we describe the network in terms of specific molecular events and causalities, in a formal language that allows no vagueness or ambiguity. Like a review, each of these statements may be proven true or false over time, but unlike a review, it is crystal clear what they mean. By itself, a GSM is not a digest or conclusion from literature, but it will also allow for that. Perhaps we can say that a GSM is the ideal review of a topic.

Secondly, the precise and formal representation of the molecular biology knowledge allows us to develop and use an analysis method to predict network level function from this knowledge base of molecular biology. Using this bipartite Boolean simulation, we can determine that the current understanding suffices to explain both healthy (wild-type) and sick (mutant) cell cycle progression or arrest. This is striking, considering that no model optimisation or fitting was performed, and that phenotypes are only evaluated as dead or alive - crudely, but surprisingly often accurately.

Third, we use these modelling results to evaluate our knowledge in terms of completeness and implications. It seems clear to us that the three replication cycles are independent (as they are known to be in e.g. meiosis) and - and this is the new insight - that the control network falls apart in three distinct control modules.

Taken together, we can - for the first time - evaluate the functionality of the cell cycle at the system level, from a definition purely at the level of molecular events and causalities. This is how we define a mechanistic understanding of a system.

Several conditions have been reported in which endocycling occurs suggesting DNA replication can cycle independent of mitosis and cell division. The authors describe their understanding of endocycling in the discussion of the manuscript "While missing may reflect missing knowledge, the modularity we observe seems to make perfect sense: The control network needs to be more than a sizer/timer; it must respond to the replication cycles it controls. Similarly, the replication cycles must be independent to explain ploidy shifts, multi-nucleate (or nuclei-free) cells and meiosis. Nevertheless, the regulatory mechanisms that uncouple these cycles remain largely unknown even in baker's yeast." The first two statements are true, but the last is inaccurate.

Answer: We disagree with the reviewer. The keyword here is "mechanisms". For a mechanistic understanding, we would need to be able to trace the information through molecular events and causalities from the initial input (starvation) to the output (initiation of meiosis; differential

activation of the division cycles). In contrast to regulation of the normal cell cycle, where we can follow the information at the molecular level, the understanding of meiosis is very limited. We are very far from (a mechanistic) understanding of these regulatory mechanisms even in baker's yeast. The role of CDK with B-type cyclins (i.e., which residues it phosphorylates and which effect each of those phosphorylations have on downstream reactions) would only be a small, albeit important, piece of this puzzle.

The role/mechanism of B-cyclins in inhibiting re-initiation of DNA replication is well-established, thus oscillation of these cyclins can drive endocycles, and this has phenomenon has been established clearly in the literature. Similar findings have been reported for SPB duplication and spindle assembly, both are components of the nuclear division cycle and can cycle independent from cell division.

Answer: We do not claim that this is a new concept. In fact, we refer in the paper to meiosis as one - of several - known examples of this uncoupling taking place. However, we present for the first time a support for this insight from a network perspective: We see that normal CDC control can be explained without any direct interaction (outside M-phase) between these modules. The paragraphs in the paper that discuss this read:

"The GSM contains the current mechanistic knowledge on information transfer and, hence, connections represent direct and functional connections in vivo. Conversely, a lack of connection implies that no (known) direct and/or functional connection exists. In particular, there is very little interaction between the three duplication cycles that execute DNA replication (DNA), nuclear division (ND) comprised of SPB duplication and nuclear division itself, and cell division (CD) involving budding, morphology and cytokinesis outside of mitosis. While the lack of direct mechanistic connections between the cycles may reflect missing knowledge, the GSM predicts that these cycles constitute distinct programs that can be executed independently through the appropriate adaptation of (the state of) the regulatory network. Such uncoupling has been observed in eukaryotic cells, leading to ploidy shifts, multinucleate cells, or cells without nuclei. The perhaps most prominent example of the modularity of these processes is meiosis, where a single DNA replication (pre-meiotic S-phase) is followed by two rounds of nuclear division (meiosis I + II) without any cell division²³. Hence, the CDC consists of three independent replication cycles; DNA replication, nuclear division and cell division."

"The genome-scale scope allows us to interpret missing features. The most striking observation is the lack of a single cycle: The control network falls apart into three distinct control circuits; G1/S, G2/M and M/G1, which monitor and control three distinct replication cycles: DNA replication, nuclear division and cell division. While missing connections may reflect missing knowledge, the modularity we observe seems to make perfect sense: The control network needs to be more than a sizer/timer; it must respond to the replication cycles it controls. Similarly, the replication cycles must be independent to explain ploidy shifts, multi-nucleate (or nuclei-free) cells and meiosis. Nevertheless, the regulatory mechanisms that uncouple these cycles remain largely unknown even in baker's yeast."

Moreover, in 1995, the Johnston group demonstrated that some *cdc6* mutants (I believe *cdc7* mutants as well) do not replicate DNA but move forward into mitosis and undergo a mitotic catastrophe. These findings demonstrated many years ago that mitosis can be initiated independent of DNA replication. Evidence suggests that the initiation of DNA replication is required for the DNA replication checkpoint to be activated and block the mitotic cycle. So again, the molecular mechanisms that couple/uncouple these cycles are well documented and involve the activities of CDKs and/or checkpoints.

Answer: Again, this does not give a mechanistic understanding of CDC regulation - only hints at a gene or protein with an important function in this process. So again, this information provides only one of many pieces of the puzzle towards a mechanistic understanding.

In the original review it was pointed out that even without checkpoints, these “independent” cycles remain ordered and cells are quite viable in the absence of perturbations. In response the authors state; “Most likely, this is due to the natural timing, as there is a natural delay between S-phase entry and mitotic entry (due to requirement for SPB duplication, bud growth and the sequential activation of the transcription factors needed to trigger the mitotic entry).” This explanation suggests some “natural timing” that offsets S-phase and M-phase entry, which is exactly the prevailing model... that there is a timer/sizer mechanism in the cell cycle control network that maintains ordering of events, and when perturbations occur, checkpoints maintain that ordering. Certainly, there are several reports that cascading transcription may be an essential part of this timing mechanism.

Answer: Our point is that the sizer/timer concept may be inaccurate and that certain transcriptional cascades may be dispensable for our understanding of cell division cycle control (in particular, the Clb5/6 -> Clb3/4 -> Clb1/2 cascade proposed by us and others). However, natural time requirements may explain why the lethality of specific regulatory mutants is masked under conditions where these regulators would not have time to become activated.

In the end, the field has already embraced the idea of separable cycles and have identified mechanisms that keep them connected, so I don't understand what novel findings have been presented.

Answer: We present the first GSM of a signalling system, perform the first formal analysis of the cell cycle control network at the genome-scale - without model optimisation or parametrisation - to evaluate the knowledge coverage and the implications of “missing” connections, and can present a holistic understanding of this network based on molecular mechanisms and causalities. Here, we see that the replication cycles appear to be independent (which is compatible with, but not the same as, the notion that they can be differentially regulated), and that the control network seems to fall apart into three unconnected - at the regulatory network layer - modules. At least the last notion is not widely accepted in the literature.

I understand and appreciate that the approach pushes the boundaries of modeling (especially at scale) and may have interest to the modeling field, but without some new (and validated finding) it's not clear why this work would be of interest to the yeast cell cycle field. Moreover, the volumes of data of different types used to construct the model are not available in nearly the depth or breadth in other systems (e.g. mammalian cell culture), so it is not clear that this approach will be generalizable to systems where the details of cell cycle control are more poorly understood.

Answer: We see an ever increasing speed of data accumulation, and there is no reason why the corresponding data could not be generated for mammalian systems. Here, we show the value of this high density mechanistic data, and we are convinced that it will only be a matter of time before the mammalian research fields catch up with yeast research.